# Neural Multisensory Scene Inference

**Jae Hyun Lim**[†§123], **Pedro O. Pinheiro**[1], **Negar Rostamzadeh**[1],
**Christopher Pal**[1234], **Sungjin Ahn**[‡§5]
[1]Element AI, [2]Mila, [3]Université de Montréal, [4]Polytechnique Montréal, [5]Rutgers University

## Abstract

For embodied agents to infer representations of the underlying 3D physical world they inhabit, they should efficiently combine multisensory cues from numerous trials, e.g., by looking at and touching objects. Despite its importance, multisensory 3D scene representation learning has received less attention compared to the unimodal setting. In this paper, we propose the *Generative Multisensory Network* (GMN) for learning latent representations of 3D scenes which are partially observable through multiple sensory modalities. We also introduce a novel method, called the *Amortized Product-of-Experts*, to improve the computational efficiency and the robustness to unseen combinations of modalities at test time. Experimental results demonstrate that the proposed model can efficiently infer robust modality-invariant 3D-scene representations from arbitrary combinations of modalities and perform accurate cross-modal generation. To perform this exploration, we also develop the Multisensory Embodied 3D-Scene Environment (MESE).

## 1 Introduction

Learning a world model and its representation is an effective way of solving many challenging problems in machine learning and robotics, *e.g.*, via model-based reinforcement learning (Silver et al., 2016). One characteristic aspect in learning the physical world is that it is inherently multifaceted and that we can perceive its complete characteristics only through our multisensory modalities. Thus, incorporating different physical aspects of the world via different modalities should help build a richer model and representation. One approach to learn such multisensory representations is to learn a *modality-invariant* representation as an abstract concept representation of the world. This is an idea well supported in both psychology and neuroscience. According to the *grounded cognition* perspective (Barsalou, 2008), such abstract concepts like objects and events can only be obtained through perceptual signals. For example, what represents a cup in our brain is its visual appearance, the sound it could make, the tactile sensation, etc. In neurosciences, the existence of *concept cells* (Quiroga, 2012) that responds only to a specific concept regardless of the modality sourcing the concept (*e.g.*, by showing a picture of Jennifer Aniston or listening her name) can be considered as a biological evidence of the *metamodal* brain perspective (Pascual-Leone & Hamilton, 2001; Yildirim, 2014) and the modality-invariant representation.

An unanswered question from the computational perspective (our particular interest in this paper) is how to learn such modality-invariant representation of the complex physical world (*e.g.,* 3D scenes placed with objects). We argue that it is a particularly challenging problem because the following requirements need to be satisfied for the learned world model. First, the learned representation should reflect the 3D nature of the world. Although there have been some efforts in learning multimodal representations (see Section 3), those works do not consider this fundamental 3D aspect of the physical world. Second, the learned representation should also be able to model the intrinsic stochasticity

---

[†]Work done during the internship of JHL at Element AI. [‡]Part of the work had done while SA was at Element AI. [§]Correspondence to jae.hyun.lim@umontreal.ca and sungjin.ahn@cs.rutgers.edu

of the world. Third, for the learned representation to generalize, be robust, and to be practical in many applications, the representation should be able to be inferred from experiences of any partial combinations of modalities. It should also facilitate the generative modelling of other arbitrary combinations of modalities (Yildirim, 2014), supporting the metamodal brain hypothesis – for which human evidence can be found from the phantom limb phenomenon (Ramachandran & Hirstein, 1998). Fourth, even if it is evidenced that there exists metamodal representation, there still exist modality-dependent brain regions, revealing the modal-to-metamodal hierarchical structure (Rohe & Noppeney, 2016). A learning model can also benefit from such hierarchical representation as shown by Hsu & Glass (2018). Lastly, the learning should be computationally efficient and scalable, *e.g.*, with respect to the number of possible modalities.

Motivated by the above desiderata, we propose the Generative Multisensory Network (GMN) for neural multisensory scene inference and rendering. In GMN, from an arbitrary set of source modalities we infer a 3D representation of a scene that can be queried for generation via an arbitrary target modality set, a property we call *generalized cross-modal generation*. To this end, we formalize the problem as a probabilistic latent variable model based on the Generative Query Network (Eslami et al., 2018) framework and introduce the Amortized Product-of-Experts (APoE). The prior and the posterior approximation using APoE makes the model trainable only with a small combinations of modalities, instead of the entire combination set. The APoE also resolves the inherent space complexity problem of the traditional Product-of-Experts model and also improves computation efficiency. As a result, the APoE allows the model to learn from a large number of modalities without tight coupling among the modalities, a desired property in many applications such as Cloud Robotics (Saha & Dasgupta, 2018) and Federated Learning (Konečný et al., 2016). In addition, with the APoE the modal-to-metamodal hierarchical structure is easily obtained. In experiments, we show the above properties of the proposed model on 3D scenes with blocks of various shapes and colors along with a human-like hand.

The contributions of the paper are as follows: (i) We introduce a formalization of modality-invariant multisensory 3D representation learning using a generative query network model and propose the Generative Multisensory Network (GMN)[1]. (ii) We introduce the Amortized Product-of-Experts network that allows for generalized cross-modal generation while resolving the problems in the GQN and traditional Product-of-Experts. (iii) Our model is the first to extend multisensory representation learning to 3D scene understanding with human-like sensory modalities (such as haptic information) and cross-modal generation. (iv) We also develop the Multisensory Embodied 3D-Scene Environment (MESE) used to develop and test the model.

## 2 Neural Multisensory Scene Inference

### 2.1 Problem Description

Our goal is to understand 3D scenes by learning a *metamodal* representation of the scene through the interaction of multiple sensory modalities such as vision, haptics, and auditory inputs. In particular, motivated by human multisensory processing (Deneve & Pouget, 2004; Shams & Seitz, 2008; Murray & Wallace, 2011), we consider a setting where the model infers a scene from experiences of a set of modalities and then to generate another set of modalities given a query for the generation. For example, we can experience a 3D scene where a cup is on a table only by touching or grabbing it from some hand poses and ask if we can visually imagine the appearance of the cup from an arbitrary query viewpoint (see Fig. 1). We begin this section with a formal definition of this problem.

A multisensory scene, simply a scene, $S$ consists of context $C$ and observation $O$. Given the set of all available modalities $\mathcal{M}$, the context and observation in a scene are obtained through the context modalities $\mathcal{M}_c(S) \subset \mathcal{M}$ and the observation modalities $\mathcal{M}_o(S) \subset \mathcal{M}$, respectively. In the following, we omit the scene index $S$ when the meaning is clear without it. Note that $\mathcal{M}_c$ and $\mathcal{M}_o$ are arbitrary subsets of $\mathcal{M}$ including the cases $\mathcal{M}_o \cap \mathcal{M}_c = \emptyset$, $\mathcal{M}_o = \mathcal{M}_c$, and $\mathcal{M}_o \cup \mathcal{M}_c \subsetneq \mathcal{M}$. We also use $\mathcal{M}_S$ to denote all modalities available in a scene, $\mathcal{M}_o(S) \cup \mathcal{M}_c(S)$.

The context and observation consist of sets of experience trials represented as *query(**v**)-sense(**x**)* pairs, *i.e.*, $C = \{(\mathbf{v}_n, \mathbf{x}_n)\}_{n=1}^{N_c}$ and $O = \{(\mathbf{v}_n, \mathbf{x}_n)\}_{n=1}^{N_o}$. For convenience, we denote the set of queries and senses in observation by $V$ and $X$, respectively, *i.e.*, $O = (V, X)$. Each query

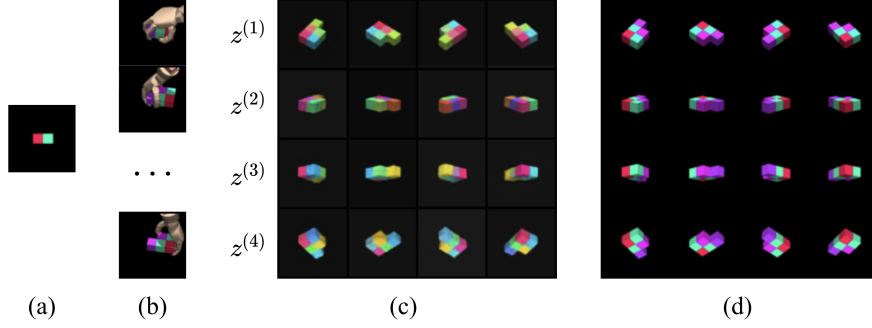

(a)      (b)            (c)           (d)

Figure 1: Cross-modal inference using scene representation. (a) A single image context. (b) Haptic contexts. (c) Generated images for some viewpoints (image queries) in the scene, given the contexts. (d) Ground truth images for the same queries. Conditioning on an image context and multiple haptic contexts, modality-agnostic latent scene representation, $z$, is inferred. Given sampled $z$s, images are generated using various queries; in (c), each row corresponds to the same latent sample. Note that the shapes of predicted objects are consistent given different samples $z^{(i)}$, while color pattern of the object changes except the parts seen by the image context (a).

$\mathbf{v}_n$ and sense $\mathbf{x}_n$ in a context consists of *modality-wise* queries and senses corresponding to each modality in the context modalities, *i.e.*, $(\mathbf{v}_n, \mathbf{x}_n) = \{(\mathbf{v}_n^m, \mathbf{x}_n^m)\}_{m \in \mathcal{M}_c}$ (See Fig. S1). Similarly, the query and the sense in observation $O$ is constrained to have only the observation modalities $\mathcal{M}_o$. For example, for modality $m = \texttt{vision}$, an unimodal query $\mathbf{v}_n^{\texttt{vision}}$ can be the viewpoint and the sense $\mathbf{x}_n^{\texttt{vision}}$ is the observation image obtained from the query viewpoint. Similarly, for $m = \texttt{haptics}$, an unimodal query $\mathbf{v}_n^{\texttt{haptics}}$ can be the hand position, and the sense $\mathbf{x}_n^{\texttt{haptics}}$ is the tactile and pressure senses obtained by a grab from the query hand pose. For a scene, we may have $\mathcal{M}_c = \{\texttt{haptics}, \texttt{auditory}\}$ and $\mathcal{M}_o = \{\texttt{vision}, \texttt{auditory}\}$. For convenience, we also introduce the following notations. We denote the context corresponding only to a particular modality $m$ by $C_m = \{(\mathbf{v}_n^m, \mathbf{x}_n^m)\}_{n=1}^{N_c^m}$ such that $N_c = \sum_m N_c^m$ and $C = \{C_m\}_{m \in \mathcal{M}_c}$. Similarly, $O_m$, $X_m$ and $V_m$ are used to denote modality $m$ part of $O$, $X$, and $V$, respectively.

Given the above definitions, we formalize the problem as learning a generative model of a scene that can generate senses $X$ corresponding to queries $V$ of a set of modalities, provided a context $C$ from other arbitrary modalities. Given scenes from the scene distribution $(O, C) \sim P(S)$, our training objective is to maximize $\mathbb{E}_{(O,C) \sim P(S)}[\log P_\theta(X|V,C)]$, where $\theta$ is the model parameters to be learned.

## 2.2 Generative Process

We formulate this problem as a probabilistic latent variable model where we introduce the latent metamodal scene representation $\mathbf{z}$ from a conditional prior $P_\theta(\mathbf{z}|C)$. The joint distribution of the generative process becomes:

$$P_\theta(X, \mathbf{z}|V, C) = P_\theta(X|V, \mathbf{z})P_\theta(\mathbf{z}|C)$$
$$= \prod_{n=1}^{N_o} P_\theta(\mathbf{x}_n|\mathbf{v}_n, \mathbf{z})P_\theta(\mathbf{z}|C) = \prod_{n=1}^{N_o} \prod_{m \in \mathcal{M}_o} P_{\theta_m}(\mathbf{x}_n^m|\mathbf{v}_n^m, \mathbf{z})P_\theta(\mathbf{z}|C). \quad (1)$$

## 2.3 Prior for Multisensory Context

As the prior $P_\theta(\mathbf{z}|C)$ is conditioned on the context, we need an encoding mechanism of the context to obtain $\mathbf{z}$. A simple way to do this is to follow the Generative Query Network (GQN) (Eslami et al., 2018) approach: each context query-sense pair $(\mathbf{v}_n, \mathbf{x}_n)$ is encoded to $\mathbf{r}_n = f_{\text{enc}}(\mathbf{v}_n, \mathbf{x}_n)$ and summed (or averaged) to obtain permutation-invariant context representation $\mathbf{r} = \sum_n \mathbf{r}_n$. A ConvDRAW module (Gregor et al., 2016) is then used to sample $\mathbf{z}$ from $\mathbf{r}$.

In the multisensory setting, however, this approach cannot be directly adopted due to a few challenges. First, unlike GQN the sense and query of each sensor modality has different structure, and thus we

cannot have a single and shared context encoder that deals with all the modalities. In our model, we therefore introduce a modality encoder $\mathbf{r}^m = \sum_{(\mathbf{x},\mathbf{v}) \in C_m} f_{\text{enc}}^m(\mathbf{x}, \mathbf{v})$ for each $m \in \mathcal{M}$.

The second challenge stems from the fact that we want our model capable of generating from any context modality set $\mathcal{M}_c(S)$ to any observation modality set $\mathcal{M}_o(S)$ – a property we call *generalized cross-modal generation* (GCG). However, at test time we do not know which sensory modal combinations will be given as a context and a target to generate. This hence requires collecting a training data that contains all possible combinations of context-observation modalities $\mathcal{M}^*$. This equals the Cartesian product of $\mathcal{M}$'s powersets, *i.e.*, $\mathcal{M}^* = Power(\mathcal{M}) \times Power(\mathcal{M})$. This is a very expensive requirement as $|\mathcal{M}^*|$ increases exponentially with respect to the number of modalities[2] $|\mathcal{M}|$.

Although one might consider dropping-out random modalities during training to achieve the generalized cross-modal generation, this still assumes the availability of the full modalities from which to drop off some modalities. Also, it is unrealistic to assume that we always have access to the full modalities; to learn, we humans do not need to touch everything we see. Therefore, it is important to make the model *learnable only with a small subset of all possible modality combinations while still achieving the GCG property*. We call this the *missing-modality* problem.

To this end, we can model the conditional prior as a Product-of-Experts (PoE) network (Hinton, 2002) with one expert per sensory modality parameterized by $\theta_m$. That is, $P(\mathbf{z}|C) \propto \prod_{m \in \mathcal{M}_c} P_{\theta_m}(\mathbf{z}|C_m)$. While this could achieve our goal at the functional level, it comes at a computational cost of increased space and time complexity w.r.t. the number of modalities. This is particularly problematic when we want to employ diverse sensory modalities (as in, *e.g.*, robotics) or if each expert has to be a powerful (hence expensive both in computation and storage) model like the 3D scene inference task (Eslami et al., 2018), where it is necessary to use the powerful ConvDraw network to represent the complex 3D scene.

## 2.4 Amortized Product-of-Experts as Metamodal Representation

To deal with the limitations of PoE, we introduce the *Amortized Product-of-Experts* (APoE). For each modality $m \in \mathcal{M}_c$, we first obtain modality-level representation $\mathbf{r}^m$ using the modal-encoder. Note that this modal-encoder is a much lighter module than the full ConvDraw network. Then, each modal-encoding $\mathbf{r}^m$ along with its modality-id $m$ is fed into the expert-amortizer $P_\psi(\mathbf{z}|\mathbf{r}^m, m)$ that is shared across all modal experts through shared parameter $\psi$. In our case, this is implemented as a ConvDraw module (see Appendix B for the implementation details). We can write the APoE prior as follows:

$$P(\mathbf{z}|C) = \prod_{m \in \mathcal{M}_c} P_\psi(\mathbf{z}|\mathbf{r}^m, m) \,. \tag{2}$$

We can extend this further to obtain a hierarchical representation model by treating $\mathbf{r}^m$ as a latent variable:

$$P(\mathbf{z}, \{\mathbf{r}^m\}|C) \propto \prod_{m \in \mathcal{M}_c} P_\psi(\mathbf{z}|\mathbf{r}^m, m) P_{\theta_m}(\mathbf{r}^m|C_m) \,,$$

where $\mathbf{r}^m$ is modality-level representation and $\mathbf{z}$ is metamodal representation. Although we can train this hierarchical model with reparameterization trick and Monte Carlo sampling, for simplicity in our experiments we use deterministic function for $P_{\theta_m}(\mathbf{r}^m|C_m) = \delta[\mathbf{r}^m = f_{\theta_m}(C_m)]$ where $\delta$ is a dirac delta function. In this hierarchical version, the generative process becomes:

$$P(X, \mathbf{z}, \{\mathbf{r}^m\}|V, C) = P_\theta(X|V, \mathbf{z}) \prod_{m \in \mathcal{M}_c} P_\psi(\mathbf{z}|\mathbf{r}^m, m) P_{\theta_m}(\mathbf{r}^m|C_m) \,. \tag{3}$$

An illustration of the generative process is provided in Fig.S2 (b), on the Appendix. From the perspective of cognitive psychology, the APoE model can be considered as a computational model of the *metamodal* brain hypothesis (Pascual-Leone & Hamilton, 2001), which states the existence of metamodal brain area (the expert-of-experts in our case) which perform a specific function not specific to input sensory modalities.

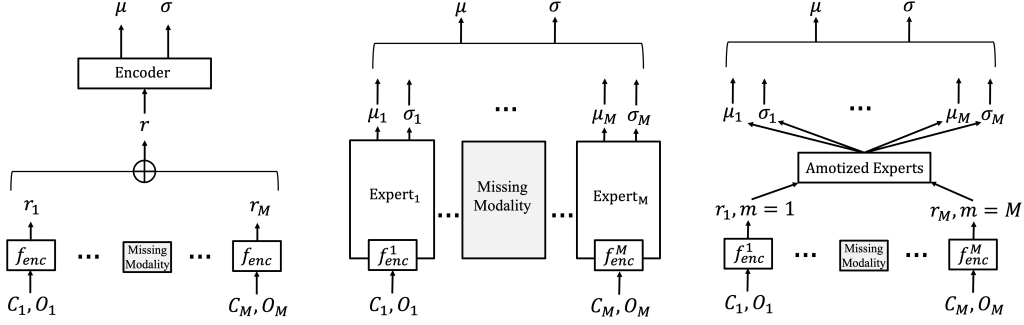

Figure 2: Baseline model, PoE and APoE. In the baseline model (left), a single inference network (denoted as *Encoder*) get an input as sum of all modality encoders's outputs. In PoE (middle), each of the experts contains an integrated network combining the modality encoder and a complex inference network like ConvDraw, resulting in $O(|M|)$ space cost of inference networks. In APoE (right), the modality encoding and the inference network are detached, and the inference networks are integrated into a single amortized expert inference network serving for all experts. Thus, the space cost of inference networks reduces to $O(1)$.

## 2.5 Inference

Since the optimization of the aforementioned objective is intractable, we perform variational inference by maximizing the following evidence lower bound (ELBO), $\mathcal{L}_S(\theta, \phi)$, with the reparameterization trick (Kingma & Welling, 2013; Rezende et al., 2014):

$$\log P_\theta(X|V, C) \geq \mathbb{E}_{Q_\phi(\mathbf{z}|C, O)}\left[\log P_\theta(X|V, \mathbf{z})\right] - \mathbb{KL}[Q_\phi(\mathbf{z}|C, O)||P_\theta(\mathbf{z}|C)] , \qquad (4)$$

where $P_\theta(X|V, \mathbf{z}) = \prod_{n=1}^{N_o}\prod_{m \in \mathcal{M}_o} P_{\theta_m}(\mathbf{x}_n^m|\mathbf{z}, \mathbf{v}_n^m)$. This can be considered a cognitively-plausible objective as, according to the "grounded cognition" perspective (Barsalou, 2008), the modality-invariant representation of an abstract concept, $\mathbf{z}$, can never be fully modality-independent.

**APoE Approximate Posterior.** The approximate posterior $Q_\phi(\mathbf{z}|C, O)$ is implemented as follows. Following Wu & Goodman (2018), we first represent the true posterior as $P(\mathbf{z}|C, O) =$

$$\frac{P(O, C|\mathbf{z})P(\mathbf{z})}{P(O, C)} = \frac{P(\mathbf{z})}{P(C, O)}\prod_{m \in \mathcal{M}_S} P(C_m, O_m|\mathbf{z}) = \frac{P(\mathbf{z})}{P(C, O)}\prod_{m \in \mathcal{M}_S}\frac{P(\mathbf{z}|C_m, O_m)P(C_m, O_m)}{P(\mathbf{z})}.$$

After ignoring the terms that are not a function of $\mathbf{z}$, we obtain $P(\mathbf{z}|C, O) \propto \frac{\prod_{m \in \mathcal{M}_S} P(\mathbf{z}|C_m, O_m)}{\prod_{i=1}^{|\mathcal{M}_S|-1} P(\mathbf{z})}$.

Replacing the numerator terms with an approximation $P(\mathbf{z}|C_m, O_m) \approx Q(\mathbf{z}|C_m, O_m)P(\mathbf{z})^{\frac{|\mathcal{M}_S|-1}{|\mathcal{M}_S|}}$, we can remove the priors in the denominator and obtain the following APoE approximate posterior:

$$P(\mathbf{z}|C, O) \approx \prod_{m \in \mathcal{M}_S} Q_\phi(\mathbf{z}|C_m, O_m) . \qquad (5)$$

Although the above product is intractable in general, a closed form solution exists if each expert is a Gaussian (Wu & Goodman, 2018). The mean $\mu$ and covariance $T$ of the APoE are, respectively, $\mu = (\sum_m \mu_m U_m)(\sum_m U_m)^{-1}$ and $T = (\sum_m U_m)^{-1}$, where $\mu_m$ and $U_m$ are the mean and the inverse of the covariance of each expert. The posterior APoE $Q_\phi(\mathbf{z}|C_m, O_m)$ is implemented first by encoding $\mathbf{r}^m = f_{\text{enc}}^m(C_m, O_m)$ and then putting $\mathbf{r}^m$ and modality-id $m$ into the amortized expert $Q_\phi(\mathbf{z}|\mathbf{r}^m, m)$, which is a ConvDraw module in our implementation. The amortized expert outputs $\mu_m$ and $U_m$ for $m \in \mathcal{M}_S$ while sharing the variational parameter $\phi$ across the modality-experts. Fig. 2 compares the inference network architectures of CGQN, PoE, and APoE.

## 3 Related Works

**Multimodal Generative Models.** Multimodal data are associated with many interesting learning problems, *e.g.* cross-modal inference, zero-shot learning or weakly-supervised learning. Regarding these, latent variable models have provided effective solutions: from a model with global latent variable shared among all modalities (Suzuki et al., 2016) to hierarchical latent structures (Hsu &

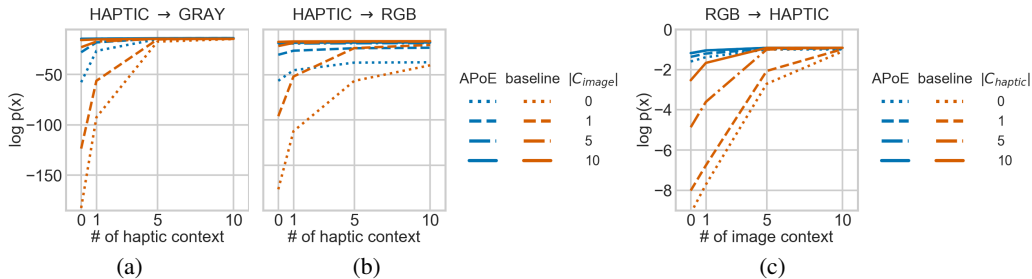

Figure 3: Results on cross-modal density estimation. (a) log-likelihood of target images (gray) vs. the number of haptic observation. (b) log-likelihood of target images (rgb) vs. the number of haptic observation. (c) log-likelihood of target haptic values vs. the number of image observations. The dotted lines show fully cross-modal inference where the context does not include any target modality. For the inference with additional context from the target modality, the results are denoted as dashed, dashdot, and solid lines.

Glass, 2018) and scalable inference networks with Product-of-Experts (PoE) (Hinton, 2002; Wu & Goodman, 2018; Kurle et al., 2018). In contrast to these works, the current study addresses two additional challenges. First, this work aims at achieving the any-modal to any-modal conditional inference regardless of modality configurations during training: it targets on generalization under distribution shifts at test time. On the other hand, the previous studies assume to have full modality configurations in training even when missing modality configuration at test time is addressed. Second, the proposed model considers each source of information to be rather partially observable, while each modality-specific data has been treated as fully observable. As a result, the modality-agnostic meta-modal representation is inferred from modality-specific representations, each of which is integrated from a set of partially observable inputs.

**3D Representations and Rendering.** Learning representations of 3D scenes or environments with partially observable inputs has been addressed by supervised learning (Choy et al., 2016; Wu et al., 2017; Shin et al., 2018; Mescheder et al., 2018), latent variable models (Eslami et al., 2018; Rosenbaum et al., 2018; Kumar et al., 2018), and generative adversarial networks (Wu et al., 2016; Rajeswar et al., 2019; Nguyen-Phuoc et al., 2019). The GAN-based approaches exploited domain-specific functions, *e.g.* 3D representations, 3D-to-2D projection, and 3D rotations. Thus, it is hard to apply to non-visual modalities whose underlying transformations are unknown. On the other hand, neural latent variable models for random processes (Eslami et al., 2018; Rosenbaum et al., 2018; Kumar et al., 2018; Garnelo et al., 2018a;b; Le et al., 2018; Kim et al., 2019) has dealt with more generalized settings and studied on order-invariant inference. However, these studies focus on single modality cases, so they are contrasted from our method, addressing a new problem setting where qualitatively different information sources are available for learning the scene representations.

## 4 Experiment

The proposed model is evaluated with respect to the following criteria: (i) cross-modal density estimation in terms of log-likelihood, (ii) ability to perform cross-modal sample generation, (iii) quality of learned representation by applying it to a downstream classification task, (iv) robustness to the missing-modality problem, and (v) space and computational cost.

To evaluate our model we have developed an environment, the Multisensory Embodied 3D-Scene Environment (MESE). MESE integrates MuJoCo (Todorov et al., 2012), MuJoCo HAPTIX (Kumar & Todorov, 2015), and the OpenAI gym (Brockman et al., 2016) for 3D scene understanding through multisensory interactions. In particular, from MuJoCo HAPTIX the Johns Hopkins Modular Prosthetic Limb (MPL) (Johannes et al., 2011) is used. The resulting MESE, equipped with vision and proprioceptive sensors, makes itself particularly suitable for tasks related to human-like embodied multisensory learning. In our experiments, the visual input is $64 \times 64$ RGB image and the haptic input is 132-dimension consisting of the hand pose and touch senses. Our main task is similar to the Shepard-Metzler object experiments used in Eslami et al. (2018) but extends it with the MPL hand.

As a baseline model, we use a GQN variant (Kumar et al., 2018) (discussed in Section 2.3). In this model, following GQN, the representations from different modalities are summed and then given to a ConvDraw network. We also provide a comparison to PoE version of the model in terms

of computation speed and memory footprint. For more details on the experimental environments, implementations, and settings, refer to Appendix A.

**Cross-Modal Density Estimation.** Our first evaluation is the cross-modal conditional density estimation. For this, we estimate the conditional log-likelihood $\log P(X|V,C)$ for $\mathcal{M} = \{\texttt{RGB-image}, \texttt{haptics}\}$, *i.e.* $|\mathcal{M}| = 2$. During training, we use both modalities for each sampled scene and use 0 to 15 randomly sampled context query-sense pairs for each modality. At test time, we provide uni-modal context from one modality and generate the other.

Fig. 3 shows results on 3 different experiments: (a) HAPTIC→GRAY, (b) HAPTIC→RGB and (c) RGB→HAPTIC. Note that we include HAPTIC→GRAY – although GRAY images are not used during training – to analyze the effect of color in haptic-to-image generation. The APoE and the baseline are plotted in blue and orange, respectively. In all cases our model (blue) outperforms the baseline (orange). This gap is even larger when the model is provided limited amount of context information, suggesting that the baseline requires more context to improve the representation. Specifically, in the fully cross modal setting where the context does not include any target modality (the dotted lines), the gap is largest. We believe that our model can better leverage modal-invariant representations from one modality to another. Also, when we provide additional context from the target modality (dashed, dashdot, solid lines), we still see that our model outperforms the baseline. This implies that our models can successfully incorporate information from different modalities without interfering each other. Furthermore, from Fig. 3(a) and (b), we observe that haptic information captures only shapes: the prediction in RGB has lower likelihood without any image in the context. However, for the GRAY image in (a), the likelihood approaches near the upper bound.

**Cross-Modal Generation.** We now qualitatively evaluate the ability for cross-generation. Fig. 1 shows samples of our cross-modal generation for various query viewpoints. Here, we condition the model on 15 haptic context signal but provide only a single image. We note that the single image provides limited color information about the object, namely, red and cyan are part of the object and almost no information about the shape. We can see that the model is able to almost perfectly infer the shape of the object. However, it fails to predict the correct colors (Fig. 1(c)) which is expected due to the limited visual information provided. Interestingly, the object part for which the context image provides color information has correct colors, while other parts have random colors for different samples, showing that the model captures the uncertainty in **z**. Additional results provided in the Appendix D suggest further that: (i) our model gradually aggregates numerous evidences to improve predictions (Fig. S5) and (ii) our model successfully integrates distinctive multisensory information in their inference (Fig. S6).

**Classification.** To further evaluate the quality of the modality-invariant scene representations, we test on a downstream classification task. We randomly sample 10 scenes and from each scene we prepare a held-out query-sense pairs to use as the input to the classifier. The models are then asked to classify which scene (1 out of 10) a given query-sense pair belongs to. We use Eq. (6) for this classification. To see how the provided multi-modal context contributes to obtaining useful representation for this task, we test the following three context configurations: (i) image-query pairs only ($I$), (ii) haptic-query pairs only ($H$), and (iii) all sensory contexts ($H + I$).

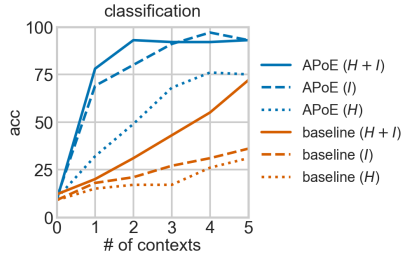

Figure 4: Classification result.

In Fig. 4, both models use contexts to classify scenes and their performance improves as the number of contexts increases. APoE outperforms the baseline in the classification accuracy, while both methods have similar ELBO (see Fig. S4). This suggests that the representation of our model tends to be more discriminative than that of the baseline. In APoE, the results with individual modality ($I$ or $H$) are close to the one with all modalities ($H + I$). The drop in performance with only haptic-query pairs ($H$) is due to the fact that certain samples might have same shape, but different colors. On the other hand, the baseline shows worse performance when inferring modality-invariant representation with single sensory modality, especially for images. This demonstrates that the APoE model helps learning better representations for both modality-specific ($I$ and $H$) and modality-invariant tasks ($H + I$).

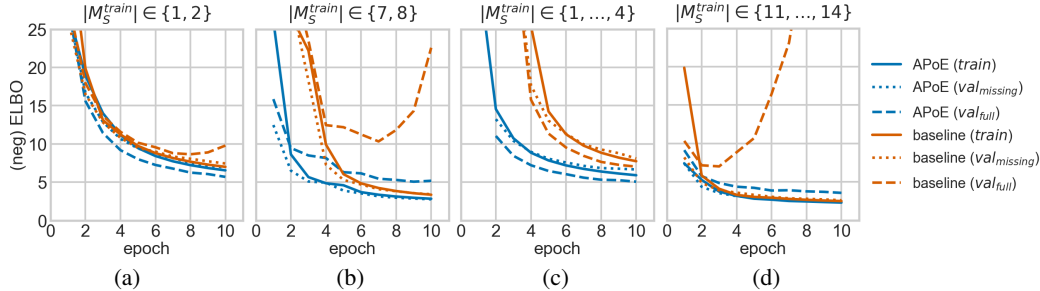

Figure 5: Results of missing-modality experiments for (a,b) $|\mathcal{M}| = 8$, and (c,d) 14 environments. During training (train), limited combinations of all possible modalities are presented to the model. The size of exposed multimodal senses per scene is denoted as $|\mathcal{M}_S^{\tt train}|$. For validation dataset, the models are evaluated with the same limited combinations as done in training ($\tt val_{missing}$), as well as all combinations ($\tt val_{full}$).

**Missing-modality Problem.** In practical scenarios, since it is difficult to assume that we always have access to all modalities, it is important to make the model learn when some modalities are missing. Here, we evaluate this robustness by providing unseen combinations of modalities at test time. This is done by limiting the set of modality combinations observed during training. That is, we provide only a subset of modality combinations for each scene $S$, *i.e*, $\mathcal{M}_S^{\tt train} \subset \mathcal{M}$. At test time, the model is evaluated on every combinations of all modalities $\mathcal{M}$ thus including the settings not observed during training. As an example, for total 8 modalities $\mathcal{M} = \{\tt left, right[3]\} \times \{R, G, B\} \times \{haptics_1\} \times \{haptics_2\}$, we use $|\mathcal{M}_S^{\tt train}| \in \{1, 2\}$ to indicate that each scene in training data contains only one or two modalities. Fig. 5 (a) and (b) show results with $|\mathcal{M}| = 8$ while (c) and (d) with $|\mathcal{M}| = 14$.

Fig. 5 (a) and (c) are results when a much more restricted number of modalities are available during training: 2 out of 8 and 4 out of 14, respectively. At test time, however, all combinations of modalities are used. We denote the performance on the full configurations by $\tt val_{full}$ and on the limited modality configurations used during training by $\tt val_{missing}$. Fig. 5 (b) and (d) show the opposite setting where, during training, a large number of modalities (e.g., 7~8 modalities) are always provided together for each scene. Thus, the scenes have not trained on small modalities such as only one or two modalities but we tested on this configurations at test time to see its ability to learn to perform the generalized cross-modal generation. For more results, see Appendix E.

Overall, for all cases our model shows good test time performance on the unseen context modality configurations whereas the baseline model mostly overfits (except (c)) severely or converges slowly. This is because, in the baseline model, the sum representation on the unseen context configuration is likely to be also unseen at test time and thus overfit. In contrast, our model as a PoE is robust to this problem as all experts agree to make a similar representation. The baseline results for case (c) seem less prone to this problem but converged much slowly. As it converges slowly, we believe that it might still overfit in the end with a longer run.

**Space and Time Complexity.** The expert amortizer of APoE significantly reduces the inherent space problem of PoE while it still requires separate modality encoders. Specifically, in our experiments, for the $\mathcal{M} = 5$ case, PoE requires 53M parameters while APoE uses 29M. For $\mathcal{M} = 14$, PoE uses 131M parameters while APoE used only 51M. We also observed a reduction in computation time by using APoE. For $\mathcal{M} = 5$ model, one iteration of PoE takes, in average, 790 ms while APoE takes 679 ms. This gap becomes more significant for $\mathcal{M} = 14$ where PoE takes 2059 ms while APoE takes 1189 ms. This is partly due to the number of parameters. Moreover, unlike PoE, APoE can parallelize its encoder computation via convolution. For more results, see Table 1 in Appendix.

## 5 Conclusion

We propose the *Generative Multisensory Network* (GMN) for understanding 3D scenes via modality-invariant representation learning. In GMN, we introduce the *Amortized Product-of-Experts* (APoE) in order to deal with the problem of missing-modalities while resolving the space complexity problem of standard Product-of-Experts. In experiments on 3D scenes with blocks of different shapes and a human-like hand, we show that GMN can generate any modality from any context configurations. We also show that the model with APoE learns better modality-agnostic representations, as well as

modality-specific ones. To the best of our knowledge this is the first exploration of multisensory representation learning with vision and haptics for generating 3D objects. Furthermore, we have developed a novel multisensory simulation environment, called the Multisensory Embodied 3D-Scene Environment (MESE), that is critical to performing these experiments.

**Acknowledgments**

JL would like to thank Chin-Wei Huang, Shawn Tan, Tatjana Chavdarova, Arantxa Casanova, Ankesh Anand, and Evan Racah for helpful comments and advice. SA thanks Kakao Brain, the Center for Super Intelligence (CSI), and Element AI for their support. CP also thanks NSERC and PROMPT.

## Footnotes

[1]Code is available at: https://github.com/lim0606/pytorch-generative-multisensory-network

[2]The number of modalities or sensory input sources can be very large depending on the application. Even in the case of 'human-like' embodied learning, it is not only, vision, haptics, auditory, etc. For example, given a robotic hand, the context input sources can be only a part of the hand, *e.g.*, some parts of some fingers, from which we humans can imagine the senses of other parts.

[3]left and right half of an image

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
