[Supplementary Material]

# A Experiments

We start from describing the Multisensory Embodied 3D-Scene Environment (MESE) environment and simulated datasets used in our experiments. We continue by explaining training settings.

$$m = \text{vision}$$

$$(x_1^m, v_1^m) \qquad (x_n^m, v_n^m) \qquad \left(x_{N_c^m}^m v_{N_c^m}^m\right)$$

$$m = \text{haptic}$$

$$(x_1^m, v_1^m) \qquad (x_n^m, v_n^m) \qquad \left(x_{N_c^m}^m v_{N_c^m}^m\right)$$

Figure S1: Example multisensory scene (with single object) in MESE. The scene includes a set of visual and haptic observations, each of which is partially observable.

## A.1 Multisensory Embodied 3D-Scene Environment (MESE)

Targeting on a development environment for 3D scene understanding through interaction, we build a multisensory 3D scene environment, equipped with visual and proprioceptive (haptic) sensors, called Multisensory Embodied 3D-Scene Environment (MESE). MESE is similar to Shepard-Metzler object experiments used in Eslami et al. (2018), but extends it with a MPL hand model of MuJoCo HAPTIX (Kumar & Todorov, 2015). The environment uses MuJoCo (Todorov et al., 2012) and the OpenAI gym (Brockman et al., 2016).

**Scene.** Adopted from Eslami et al. (2018), MESE generates single Shepard-Metzler object with an arbitrary number of blocks per episode. Each block of the object is randomly colored in HSV scheme. More precisely, hue and saturation are randomly selected within fixed ranges: hue is sampled from (0, 1) and saturation is sampled from (0, 0.75). Value (in HSV) is fixed to 1. The sampled HSVs are converted to RGBs.

**Image.** An RGB camera is defined in the environment for visual input. The position of the camera and its facing direction, *i.e.* $(x, y, z, pitch, yaw)$ are defined as actions for agents. We refer to a viewpoint as the position and facing direction combined. From a given viewpoint, a generated RGB image has $3 \times 64 \times 64$ dimension.

**Haptic.** For proprioceptive (haptic) sense, the Johns Hopkins Modular Prosthetic Limb (MPL) (Johannes et al., 2011) is used, which is a part of MuJoCo HAPTIX. The hand model generates 132-dimensional observation, consisting of the its actuator positions, velocities, accelerations, and touch senses. For more details about the MPL hand, please finds Appendix C. in Amos et al. (2018). The MPL hand model has 13 degrees of freedom to control. MESE adds 5 degrees of freedom, *i.e.* $(x, y, z, pitch, yaw)$, to control the position and facing direction of the hand's wrist, similar to camera control.

## A.2 Datasets

Given that each scene has a single object at the origin, images and haptics are randomly generated. For an image, a camera viewpoint is sampled on a spherical surface with a fixed radius while the camera faces to the object. We refer to image query as camera viewpoint.

For an haptic data in each scene, we first sample a wrist pose of the hand similar to generating camera viewpoints. Given the sampled wrist, a fixed deterministic policy is performed.The policy starts from a stretched hand pose to gradually go to grabbing posture without any stochasticity. Note that a haptic datapoint is a function of the wrist pose and the object, given the aforementioned fixed policy; thus, the wrist's position and facing direction is set to the hand's query. Each dimension of haptic data is re-scaled to $[-1, 1]$.

For the environment $\mathcal{M} = \{\texttt{RGB-image}, \texttt{haptics}\}$, also denoted as $|\mathcal{M}| = 2$, 1M scenes are collected for training data. For each scene, a Shepard-Metzler object with 5 parts is randomly sampled as described in Section A.1. The number of unique shapes is 726 for the 5-parts object dataset. In each scene, 15 queries and corresponding sensory outputs are randomly sampled for each sensory modularity. For validation and test data, 20k and 100k scenes are sampled, respectively.

For the environment whose $|\mathcal{M}|$ is larger than 2, we slice the dimensions of image and haptic data. For example, in order to build an environment $|\mathcal{M}| = 5$, $3 \times 64 \times 64$ image is split to four quadrants of it so that each $3 \times 32 \times 32$ image is one of $\{\texttt{upper-left-RGB}, \texttt{upper-right-RGB}, \texttt{lower-left-RGB}, \texttt{lower-right-RGB}\}$. In addition to these four visual modalities, haptic input is provided. Note that while we split each image into four, the corresponding experiment defers from image in-paining or de-noising tasks. In those image tasks, statistical regularities of image are heavily taken into account, *i.e.* statistics of local receptive fields are almost identical regardless of position. Many recent solutions on the problems resort to convolutional architectures, as a practical solution for sharing parameters of models across arbitrary locations. As long as the inductive bias hasn't made use of in any model, it is valid that they are distinct random variables, each of which has different statistical characteristic; thus, they can be treated as multiple modalities.

For $|\mathcal{M}| = 8$, image is cropped to $3 \times 56 \times 56$ and re-sized to $3 \times 48 \times 48$ due to the memory overhead. The image is split to left-right for each RGB channel; thus, we have $\{\texttt{left-R}, \texttt{left-G}, \texttt{left-B}, \texttt{right-R}, \texttt{right-G}, \texttt{right-B}\}$ as different senses. Haptic dimension is also divided into to two, *i.e.* $\texttt{haptics}_1$ and $\texttt{haptics}_2$. $\texttt{haptics}_1$ corresponds to thumb, index, and middle fingers. $\texttt{haptics}_2$ corresponds to ring and little fingers, as well as palm.

For $|\mathcal{M}| = 14$, image is converted to $3 \times 48 \times 48$ as in $|\mathcal{M}| = 8$, but is is sliced as to $\{\texttt{upper}, \texttt{lower}\} \times \{\texttt{left}, \texttt{right}\} \times \{\texttt{R}, \texttt{G}, \texttt{B}\}$. With haptic data divded to $\texttt{haptics}_1$ and $\texttt{haptics}_2$, we get an environment with $|\mathcal{M}| = 14$.

### A.3 Training

For training, Adam optimizer is used (Kingma & Ba, 2014). $\beta$-annealing[4] is employed; $\beta$ is set to $0.1 \sim 1$ for the first epoch and maintained as 1 for the rest of training. Learning rate is set to 0.0001. In order for stable training, gradient is clipped to $[-0.25, 0.25]$. Training is ran for 10 epochs. Mini-batch size is set to 14 scenes for the $|\mathcal{M}| = 2$ environment and 24 scenes for the $|\mathcal{M}| = 5, 8$, and 14.

## B  Network Architectures

**Overall.** We adopt C-GQN network architecture from Kumar et al. (2018) for the proposed model, as well as the baseline. This architecture can be thought of as a modified version of ConvDraw encoder-decoder, in which the posteriors don't have feedback routes of the predicted inputs and the residuals between the target and the predictions, unlike the original one (Gregor et al., 2016). As a result, for every step of ConvLSTM iterations the same input is repeatedly provided instead (see Fig. S3 (a)).

For a baseline, we use the C-GQN network, and the baseline's generation process is depicted in Fig. S2 (a). Each instance of $m$-th modality query-sense pairs feeds to $f_{\text{enc}}^m$, *i.e.* $m$-th representation network. All instances of representation $\mathbf{r}_n^m$s will summed up to get representation $\mathbf{r}$. Metamodal scene representation $\mathbf{z}$ is inferred using the C-GQN decoder (or encoder in inference). Conditioning on the $\mathbf{z}$ and a query $\mathbf{v}_n^m$, a sensory datapoint will be generated using $g_{\text{dec}}^m$, *i.e.* the *renderer* for the $m$-th modality.

For APoE, multiple experts are modeled as a single network, called expert-amortizer, in which a binary mask to identify modularity is used while inferring $z$, e.g. $Q_{\phi_m}(\mathbf{z}|\mathbf{r}^m) = Q_\phi(\mathbf{z}|\mathbf{r}^m, m)$ for $\forall m \in \mathcal{M}$ in Eq. (2) where $m = $ a binary mask. The expert-amortizer is build upon further modifications from the modified ConvDraw, as shown in Fig. S3 (b). Especially for efficient computation, the expert-amortizers are implemented such that they perform convolution over $\mathbf{r}^m$ for $\forall m \in \mathcal{M}_S$.

See Fig. S2 (b) for APoE's generation. Identical to the baseline, each instance of $m$-th modal query-sense pairs feeds to $f_{\text{enc}}^m$, and they are summed up to get modality-specific representation $\mathbf{r}^m$. However,

(a) Baseline model.

(b) APoE.

Figure S2: Computation graphs of generation processes for the proposed models. (a) Baseline model: Each instance of $m$-th modality query-sense pairs feeds to $f_{\text{enc}}^m$, *i.e.* representation network. All instances of representation $\mathbf{r}_n^m$s will summed up to get representation $\mathbf{r}$. Metamodal scene representation $\mathbf{z}$ is inferred using the C-GQN decoder (or encoder in inference). Conditioning on the $\mathbf{z}$ and a query $\mathbf{v}_n^m$, an instance of sensory data will be generated using $g_{\text{dec}}^m$, *i.e.* the *renderer*. (b) APoE: Each instance of $m$-th modality query-sense pairs feeds to $f_{\text{enc}}^m$, and they are summed up to get modality-specific representation $\mathbf{r}^m$. Metamodal scene representation $\mathbf{z}$ is inferred via product-of-experts using the expert-amortizer network. Rendering follows the same process as in the baseline model.

Figure S3: Implementation details for the modified ConvDraw network architecture; a) baseline encoder and b) decoder. c) the proposed model's encoder and d) decoder. Sampled latent $z = [z_1, z_2, \ldots, z_T]$ will be passed to renderers. Note that in the PoE and APoE, the distribution of $z_t$ is estimated as the product of $m$ experts for each $t$-th step.

metamodal scene representation $\mathbf{z}$ is inferred via product-of-experts using the expert-amortizer network.

For PoE, each expert is modeled a single ConvDraw encoder-decoder with corresponding modularity encoder, and the rest of its implementations are identical to APoE.

**Representation Network.** To estimate modality-specitic representation for each instance of a query-sense pair, tower representation networks proposed in Eslami et al. (2018) is used. For camera position-image pair, convolution layer is used in the tower representation network. Similar to image, MLP is applied for a haptic observation and its corresponding query. The same representation network architectures are used to baseline, PoE, and APoE.

**Renderer.** Renderer network is a part of a decoder for predicting each sensory modality. Those renderers get a query $\mathbf{v}_m^n$ and modality-agnostic latent representation $z$ as its inputs, output the sensory data conditioning on them. For rendering image, ConvLSTM with convolutional layer is used as done in C-GQN. Similar to image rendering, ConvLSTMs is used for proprioceptive, but MLP is employed instead of convolution layer.

# C   Classification

For classification, we adopt a method from Lake et al. (2015). Let we have $K$-number of context sets, $C^{(k)}$ for $\forall\, k$, each of which is a set that contains multisensory data-query pairs. Given we have an observation set $O = \{X, V\}$ obtained from one of the $K$-scenes (more precisely objects), we can predict from which scene the new observation set comes. The predicted label $\hat{k}$ is obtained by following;

$$\hat{k} = \arg\max_{k} \log P_\theta(X|V, C^{(k)}), \tag{6}$$

where each $\log P_\theta(X|V, C^{(k)})$ is estimated by using the log-likelihood estimators from Burda et al. (2016). This method doesn't require additional training process. To approximate the log-likelihood each $k$, 50 latent samples are used.

For held-out dataset, 1000 additional Shepard-Metzler objects with 4 or 6 parts are generated: any of these objects hasn't presented in training dataset. $K$ is set to 10. For all models, three different inference scenarios are considered; classification is performed by using (i) only image-query pairs from each scene ($I$), (ii) haptic-only contexts ($H$), and (iii) use both sensory contexts ($H + I$).

The results are shown in S4. In order to claim that both models are well trained and converged to training dataset, the learning curves for baseline and APoE models used in classification are also attached.

Figure S4: Classification result for $|\mathcal{M}| = 2$ environment. (Left) learning curves for baseline and APoE models used in classification. (Right) classification results of the models from the left. Classification is performed in three different conditioning scenarios; each model is conditioned on image-only context ($I$), haptic-only context ($H$) and all information ($H + I$).

# D   Cross-modal Generation

## D.1   Reducing Uncertainty with Aggregation of Evidences

In this task, we examine the uncertainty of modality-agnostic representations with respect to the number of contexts. Similar to Fig. 1, we provide a single image context but we condition a trained model on different numbers of haptic contexts. More precisely, the image context is given such that the model cannot recover the entire scene from the image.

The generated image samples are shown in Fig. S5 (a). As the number of the haptic contexts increases, more accurate visual observation is predicted. We can also observe that generated images at each column continue to develop in comparison to the previous column, corresponding to where additional haptic information is provided. Again, we observe that the part of the object for which the context image provides color information has similar colors while other part of the block has random colors.

Fig. S5 (b) describes the generated haptic samples from the same query. In this figure, 95%-confidence interval from 20 samples is also illustrated. Similar to visual prediction, the haptic prediction improves according to the number of the haptic contexts. In addition, it is demonstrated that the uncertainty of the prediction reduces as the contexts aggregate.

$$
\begin{array}{cccccc}
& 0 & 1 & 1\dots2 & 1\dots3 & 1\dots4
\end{array}
$$

(a)                                   (b)

(c)

Figure S5: Multi-sensory inference. (a) Upper row illustrates single visual context and various haptic cues, from empty observation to multiple observations. Note that the given visual context is insufficient to infer correct object shape. The model predicts visual observations for different viewpoints, *i.e.* $v_1$, $v_2$, and $v_3$, using the visual and haptic contexts. $x$-axis label indicates the indices of haptic contexts used when the model predicts the corresponding column. (b) The ground truth images for the same viewpoints. (c) The model predicts haptic observations for a haptic query. The ground truth values are denoted as red. $x$-axis label indicates index of 132-dimensional output of the hand model. $|C_{\texttt{haptics}}|$ means the number of haptic contexts used for prediction.

## D.2 Any-to-any Cross-modal Generation

Additional cross-modal generation experiments are performed for $|\mathcal{M}| = 5, 8$, and, $14$ in order to explore multisensory integration of arbitrary context conditions. Given any context condition, a trained model is asked to generate all modality outputs (for a given set of queries) and these are combined to be displayed. For instance, a model trained in $|\mathcal{M}| = 14$ generates outputs in all modalities, *i.e.* $\mathcal{M} = \{\texttt{upper}, \texttt{lower}\} \times \{\texttt{left}, \texttt{right}\} \times \{\texttt{R}, \texttt{G}, \texttt{B}\} + \texttt{haptics}_1\texttt{haptics}_2$. The visual outputs are combined and displayed as shown in Fig S6 (d). The haptic outputs are omitted to conserve space.

Three different context conditions are applied for each environment. For $|\mathcal{M}| = 5$, a model is conditioned on (i) $\texttt{haptics-only}$, (ii) $\texttt{haptics+upper-left-RGB}$, and (iii) $\texttt{haptics+upper-left-RGB+}$ $\texttt{lower-right-RGB}$ contexts. For $|\mathcal{M}| = 8$, a model is conditioned on (i) $\texttt{haptics}_1 + \texttt{haptics}_2$, (ii) $\texttt{haptics}_1 + \texttt{haptics}_2 + \texttt{left-R}$, and (iii) $\texttt{haptics}_1 + \texttt{haptics}_2 + \texttt{left-R} + \texttt{right-G}$. For $|\mathcal{M}| = 14$, a model is conditioned on (i) $\texttt{haptics}_1 + \texttt{haptics}_2$, (ii) $\texttt{haptics}_1 + \texttt{haptics}_2 + \texttt{upper-left-R}$, and (iii) $\texttt{haptics}_1 + \texttt{haptics}_2 + \texttt{upper-left-R} + \texttt{lower-right-B}$. Each context modality is provided with 5 query-sense pairs.

The results are shown in Fig. S6 (b)-(d), and the ground truth images are given in Fig. S6 (a). The provided context senses are illustrated in the first and second rows in each experiment. In general, haptic-related contexts are sufficient for the learned models to infer the shapes. With additional visual cues, the models start to correctly predict colors. For example, in middle column of Fig. S6 (c) and (d), red-mixed colors are successfully inferred with $R$-channel context; however, it still fails to predict all color patterns as other color information is deficient. As more color information is given, our models results in successful predictions of all color patterns as shown in the right column of Fig. S6 (c) and (d).

## E   Missing-modality Problem

In addition to the experiments described in Fig. 5, more results are added in S7. Note that `train` loss is evaluated as moving average of mini-batches, while $\mathtt{val_{missing}}$ is estimated on whole batch at the end of each epoch. This explains validation loss sometimes lower than training's in the figures.

In general, all models tend to under-fit when they have never seen entire modalities during training. On the other hand, the models exposed to many modalities are prone to give us tighter negative ELBO.

We can observe notable difference between the baseline and APoE on the settings secluded from individual modality during training. Combined with the classification in Fig. S4, we can interpret the results as that PoE helps training individual expert. The inference of PoE has been understood as an agreement of all experts (Hinton, 2002); therefore, this lead each expert is capable of performing inference independently as well as expressing its own uncertainty. On the other hand, the simple sum operation of the C-GQN (baseline) probably end up with relying on dominating signals and ignore rests, which drove to overfit to training distributions.

## F   Computational Time

| Model | # of parameters | | | | timer per iter (ms) | | | |
|---|---|---|---|---|---|---|---|---|
| | $\mathcal{M}=2$ | 5 | 8 | 14 | $\mathcal{M}=2$ | 5 | 8 | 14 |
| baseline | 53M | 28M | 48M | 51M | 346 | 397 | 481 | 866 |
| APoE | 53M | 29M | 48M | 51M | 587 | 679 | 992 | 1189 |
| PoE | 58M | 53M | 92M | 131M | 486 | 790 | 1459 | 2059 |

Table 1: The number of parameters and computation time for each model for all experiments. Mini batch size is set to 1.

Table 1 shows the number of parameters and computational time cost for all experiments. Each experiment is ran with single NVIDIA Tesla P100 GPU and four cores of an Intel Xeon E5-2650 2.20GHz CPU. PyTorch (Paszke et al., 2017), CUDA-9.0 (Nickolls et al., 2008), and cuDNN7 (Chetlur et al., 2014) are used for the implementations. All models share the same representation and renderer network architectures, and the same number of steps and hidden sizes are applied to the encoder and decoder architectures. For fair comparison, the mini-batch size is set to 1 for measuring the costs.

In PoE, each of the expert contains a large network like ConvDraw, resulting in $O(|M|)$ space cost for inference networks. In APoE, the inference networks are integrated into single expert-amortizer, serving for all modalities. Thus, the space cost of inference networks reduces to $O(1)$. As a result, the APoE model's parameter size in the experiments is almost the same as the baseline's, while it can provide probabilistic information integration that the PoE has.

(a) Ground-truth images

(b) $|\mathcal{M}| = 5$ environment, *i.e.* $\{\texttt{upper}, \texttt{lower}\} \times \{\texttt{left}, \texttt{right}\} + \texttt{haptics}$.

(c) $|\mathcal{M}| = 8$ environment, *i.e.* $\{\texttt{left}, \texttt{right}\} \times \{\texttt{R}, \texttt{G}, \texttt{B}\} + \texttt{haptics}_1 + \texttt{haptics}_2$.

(d) $|\mathcal{M}| = 14$ environment, *i.e.* $\{\texttt{upper}, \texttt{lower}\} \times \{\texttt{left}, \texttt{right}\} \times \{\texttt{R}, \texttt{G}, \texttt{B}\} + \texttt{haptics}_1 + \texttt{haptics}_2$.

Figure S6: Any-to-any cross-modal generation examples. Given context conditions, trained models are asked to generate all modality outputs (for a given set of queries) and these are combined to be displayed. The first row in each sub-figure displays five haptic context senses. The second row illustrates combined senses from different visual modalities. The third illustrates the predictions for given queries. For example, the second row in (b) depicts the five upper-left-RGB senses. The same row in (c) displays additional five lower-right-RGB senses combined with the ones in (b).

(a) $|\mathcal{M}| = 5$ environment, *i.e.* $\{\texttt{upper}, \texttt{lower}\} \times \{\texttt{left}, \texttt{right}\} + \texttt{haptics}$.

(b) $|\mathcal{M}| = 8$ environment, *i.e.* $\{\texttt{left}, \texttt{right}\} \times \{\texttt{R}, \texttt{G}, \texttt{B}\} + \texttt{haptics}_1 + \texttt{haptics}_2$.

(c) $|\mathcal{M}| = 14$ environment, *i.e.* $\{\texttt{upper}, \texttt{lower}\} \times \{\texttt{left}, \texttt{right}\} \times \{\texttt{R}, \texttt{G}, \texttt{B}\} + \texttt{haptics}_1 + \texttt{haptics}_2$.

Figure S7: Results of missing-modality experiments for various multimodal scenarios. During training, limited combinations of modalities are presented. At test time, all combinations of the entire modalities are randomly selected.

## Footnotes

[4]In here, $\beta$-annealing refers to anneal a weight at KL term of ELBO as done in Higgins et al. (2017)