[Reviews · NeurIPS 2019]

Reviewer 1



In a nutshell, the GMN architecture aims at: * any-modal to any-modal conditional inference, regardless of any modality configurations present during training * handling partial observability in each modality and achieves these goals by constructing a multimodal, abstract scene representation in to stages: 1) observations are deterministically embedded in an input-modality specific way 2) embedding are fed (alongside a one hot indicator of the input modality) into a shared expert-amortizer (implemented with ConvDraw). The changes in the scene encoder architecture, as well as the implications on the performance of the model compared to the prior art are very significant. Figure 2 is particularly striking: comparing the baseline model (CGQN) and GMN it's possible to appreciate how the fundamental change is moving DRAW before or after the aggregation step. In the baseline the observation encodings r_i are summed component-wise first and fed to DRAW to produce a distribution over scene latents z - it is up to the modality encoders (tipically low capacity resnets/convnets) to produce embeddings that, after being summed together, can be correctly interpreted by DRAW. In GMN, r_i are passed along side the id of their corresponding modalities to DRAW, which returns a set of distribution over encodings that can be easily aggregated via component-wise sum in close form. Based solely on this description I would wonder about this tradeoff in representation power: in GMN more capacity is used to produce a representation that can be more easily aggregated, in CGQN more capacity is used to transform the aggregated signal into a distribution that could better capture multimodalities and uncertainty using all the inputs. Am I correct in this interpretation of the results? Could the authors comment on this in the text? The authors mentioned they have considered using a probabilistic observation encoder, but preferred using a deterministic one in their experiments; does this imply that they tried to merge the two approaches using a deeper, perhaps autoregressive observation encoder, but failed to train the model? The experimental section is very thorough and thought provoking, providing benchmarks on both the model performance in observation prediction, and analysis of the scene representation. The supplementary materials provide lots of detail on implementation and training/testing regimes to facilitate reproducibility. Furthermore, in order to obtain multiple observation modalities the authors generate a new dataset with both vision and haptic data which will be opensourced - this is indeed great, but I wish they had also included as 3rd modality depth data (perhaps this could be provided to the model at a lower resolution and degraded with noise to make the task a bit harder). A few presentation issues/questions: Line 57: Why does aPoE improve computational efficiency? You still have to forward pass and back propagate to the network, even if the weights are shared. Can the authors clarify? Line 60: At this point in the paper I could not follow the comments about 'modal-to-metamodal hierarchical structure': this part of the introduction is a bit confusing! Please consider commenting further to make the statement clearer (the paragraph at line 73 was helpful, but it was much later in the paper) and substantiate the claim pointing to experiments or relevant literature. Figure 3 is visually crammed and not very readable - perhaps it's the section of the paper that took me the longest to parse. The author should consider redesigning the figure with readability in mind. I found that Figure 5's caption is also very hard to parse. I found the final note on using moving averages vs not-moving averages is baffling. I think you shouldn't confuse the reader and burden them with the work of interpreting the curves when the underlying data is not consistently generated; or you should at least explain in detail what and why is unexpected. Nonetheless, the experiments in figure 5 are extremely interesting wrt to generalization considerations. Line 299: in this paragraph the authors should also included the data regarding the baseline model (which is reported in the supplementary material) - it would make the paper even stronger, since the GMN is not significantly bigger that CGQN, but consistently outperforms it in the experiments.

Reviewer 2



[Originality] The manuscript proposes a combination of GQN-style conditional latent-variable models with multi-modal generative models to be trained with variational inference. The combination is well motivated given the multi-modal nature of observations of a 3D scene, although the novelty is hard to fully specify since multi-modal generative models easily allow for conditioning through queries, but actually employing the GQN architecture is not entirely trivial either. To perform inference, the authors propose an amortised PoE variational posterior that ties together some weights in the standard PoE construction due to parameter constraints of the ConvDraw encoder used (via GQN). This claim does not seem all that novel given the main PoE formulation of the posterior for multi-modal observations exists [1]. In fact, the derivations given (l165-l169) seem to be the same as those in eqns 3 and 4 in [1]. The shared weights, and potentially distinct priors per modality (l133-l135) seem to be the only additional elements. [Quality] The formulation appears to be sound---the multi-modal conditioning model seems right. I do have one particular issue with regard to related work however---the claim in Sec 3, that previous studies have assumed full-modality configurations does not appear to be entirely true. In [1], sections 2.2 and 5.1 talk about how the PoE variational posterior can be used to deal with missing modalities just as discussed here. Unless I misunderstood the claim, this is a feature of PoE itself, and is not novel to this work or indeed [1]; just that it is an additional feature that PoE endows upon the setup. On the experimental results front, the reported results look good, but, given that the APoE is the claimed addition over PoE (which has been previously derived in [1]), it is puzzling that the authors do not include a comparison to a standard PoE model---surely this would be necessary to tell if the gains seen in the results are because of PoE or the additional amortisation? This should have not been too much trouble given that [1] have released code [2] for their work. [Clarity] The manuscript is well written, with good organisation. The experimental details and exposition could be made a little less dense, but given space constraints, is understandable. [Significance] The experimental setting is an apposite one, and timely given the recent interest in learning robust scene representations that have better utility on downstream tasks. The new environment looks interesting, and pending release, is a useful contribution to the community. As described above, this work extends GQN-style methods with multi-modality, but it's value in extending PoE variational multi-modal inference is not clear. Overall, I'm close to giving it a positive score, but for the fact that the experiments don't compare against an actual pre-existing and applicable model [1], instead comparing to a GQN baseline, and the relative work appears to mischaracterise prior work on their requirement for full-modality configurations. [1] Multimodal Generative Models for Scalable Weakly-Supervised Learning, Wu and Goodman, NeurIPS 2018 [2] https://github.com/mhw32/multimodal-vae-public **Update** I've read through the rebuttal, and I'm a little bit underwhelmed by it. I find it difficult to agree that sharing some parameters between encoders in a PoE posterior counts as a 'contribution'---this is way too obvious a thing to do when the encoders themselves are hefty; even more so when considering their own response effectively casting this as purely an computational-efficiency change. As for the argument that the PoE couldn't be run because of the practical issues, that was a predictable answer, but still something I would like to see a _concrete_ response for, not just a hand-wavy one! Going by Table 1 in the supplement, the authors could have compared for M=2,5 --- where both the number of parameters and iteration time appear comparable? For M=5, the #parameters stays mostly the same as M=2. Even for M=14, 131M parameters for the PoE suggest approximately 525Mb of memory (assuming floats) to store the model, which definitely is not so large as to be prohibitive---are there resource constraints limiting access to GPUs of 2GB, or 4GB even? And for the iteration time, for M=14, it seems only double that of the APoE, which again does not seem prohibitively large given that most runs appear to only be 10 epochs? Now, I'm not saying I expect comparisons for M=14---that is a function of what hardware resources available---but lower M should have been possible because the APoE was runnable? I'm happy with the clarifications on the abilities of Wu & Goodman---I hope the authors do make it very clear that the formulation and capabilities are derived from that work, and avoid confusion about claimed contributions (citations in the right place, etc.). Apart from the above: I think the MESE data and environment is a clear contribution. The multi-modal extension of GQN I'm unsure about as a contribution, but am willing to give the benefit of doubt to. However, my main worry with things is that the implications are being drawn, or the insights reported, against a baseline that is _not_ an apples-to-apples comparison. I'm not totally convinced that a PoE comparison is not feasible (see arguments above); at least based of the authors' response. I do believe some satisficing for the improvements I asked for, so I can upgrade my score a notch, but I'm worried the narrative is somewhat problematic as is. I would hope that the authors take the baseline comparison seriously and make the necessary edits to actually show that their conclusions are well-founded, doing the utmost to make sure that improvements claimed are purely a result of their actual contributions (APoE, not PoE).

Reviewer 3



Summary This paper proposes Generative Multisensory Network, a multisensory 3D representation learning algorithm. The authors first formalize the problem, writing it in the form of maximum log-likelihood estimation. A scene representation variable is then introduced, and the authors mention that modeling the representation variable prior by using the existing method Product-of-Experts is limited due to limited computational and storage resources. So the authors present Amortized Product-of-Experts to address this issue by using only one model, taking the modal-encoding along with its modality-id as inputs. The paper also talks about how to conduct inference on the proposed model by using some approximations. Strengths -The formulation of the problem is novel and general. As illustrated in section 2.1, it unifies different modalities so that it can be applied to 3D scene or even more general cases. -Modeling 3D scene by a generative query network is novel. 3D scenes are definitely difficult to model, and it is an interesting way to model it by a generative query network, which takes in observation parameters and generates observations. -The paper is clear. The authors write most of the contents in formulas, which are concise and clear. Weaknesses -The testing 3D scene is synthetic and simple. From my understanding, the scenes only contain a bunch of cubes of different colors. Testing on the simple and synthetic case is a good start, and it would be great if we can later test the model on real and/or complex scenes. Comments after the rebuttal ************************************ Thank authors for the rebuttal.  After reading reviews and rebuttal, I decided to lower the score to 6, as I noticed something not satisfying of this paper. 1) I first thought modeling a 3D scene by a generative query network is pretty novel, but the idea is actually from GQN paper. This is not a deficiency, but I previously mistook this as a strength of the paper; 2) The comparison between PoE and APoE in the rebuttal is not satisfying. For small |M| like 2, APoE is (almost) not saving time or space. For larger |M|, we need a performance comparison with PoE. I strongly recommend adding PoE result. That said, I still think this paper is marginally above the acceptance threshold since it has the potential to be a good paper as the authors address the concerns from the reviewers.

[Author Response · NeurIPS 2019]

Thank you for the positive, constructive and in-depth reviews. We found the suggestions and comments to be very
helpful. Below, we summarize the main questions and comments raised by each reviewer and provide responses.

**(REVIEWER 1) Trade-off in representation power.** The CGQN representation does not model agreement among
the individual modalities as long as the summed representation can model the scene. Thus, it suffers if an unseen
combination is given at test time. In contrast, the PoE and APoE approaches are robust to this issue as they seek the
agreement through the product of experts operation. Being free from this agreement constraint, CGQN representations
may be obtained by searching a broader model space than PoE-based models. However, our experiments show that
having this agreement constraint (so searching the representation from a less broad space) does not hurt the scene
modeling performance while solving the partial observability problem.

**Probabilistic hierarchical encoder.** We have not tried the hierarchical latent model version. Although this is an
interesting model variant, due to space limitation we didn't prioritize exploring it. We believe that it will not be difficult
to train that model as the reparameterization trick can be applied there as well. However, evaluating the advantages of
having the hierarchical probabilistic encoding would require more thought.

**3rd modality depth data.** We agree and hope to explore this in the future work.

**Why does APoE improve computation efficiency?** In principle, APoE should have the same computation complexity
as PoE. However, in practice we observe that our amortization helps improve the computation speed as well. It seems
that the amortized ConvDraw uses GPU parallel computation more efficiently that having a ConvDraw for each modality.
For better understanding, we need a more investigation on how PyTorch and CUDA actually realizes this efficiency.

**Scene representation for RL.** We agree that extending this work in a real world setting with a robot arm and RL policy
learning is a very interesting future direction.

**Other comments on the presentation.** We agree on all the points and will improve it in our camera-ready.

**(REVIEWER 2) In comparison to Wu & Goodman (2018)** We agree that, our approach to addressing the partial
observability problem extends the work of Wu and Goodman (2018) by adopting the PoE modeling approach (we
will make this point clearer). However, we do not claim this as our contribution. We see our main contributions as
being: (i) the formalization and demonstration of *3D modality-invariant representation learning and generation using*
*human-like multisensory inputs under the GQN framework*, (ii) the introduction of an amortized PoE which resolves
the problem that the Wu and Goodman model suffers from when it is applied to our 3D modeling problem, namely
the inherent scalability problem of the PoE model due to space complexity, and (iii) the introduction of the MESE
simulation environment which we believe is a significant contribution, given that there is no such environment currently
available to the community. In addition, we agree that it is worth noting that the derivation of lines 165-169 is from Wu
& Goodman (2018). We cite them on line 171, but we will make this point clearer.

**Experiments on PoE by Wu & Goodman (2018)** We indeed tried to run the PoE model of Wu and Goodman in
our problem by integrating it with a CGQN. (Note that the MVAE model of Wu and Goodman cannot directly be
used for our 3D modeling problem.) However, we could not run it properly after the CGQN integration due to its
increased memory footprint and slowed speed. This is how we arrived at our position that the amortization contribution
is indeed important. Also, in terms of accuracy, we do not think our APoE will be better than PoE. The PoE should be
the upper bound of our model because our model is an amortized version of it, i.e., due to the amortization gap. So,
the advantage of amortization in our case is mainly on improving efficiency and scalability not necessarily on better
accuracy. Considering both reasons, we believe that it is fine not to compare to PoE in terms of accuracy—but we
provided the comparison in terms of space and computation efficiency. If reviewers think it will nevertheless be helpful,
we will be happy to find a way to add the PoE result in the camera-ready version.

**Full-modality configuration in previous work.** We fully agree on this point. There was some confusion because the
way we implement the missing modality situation is somewhat more challenging in the sense that we also consider the
case where the fully joint modalities are not available at all. We, however, agree that this is not a significant difference
as currently described in the related work section. We will fix the description and clarify that this problem is studied in
Wu & Goodman (2018).

**(REVIEWER 3)** Thanks for the constructive review. **More complex task.** We agree on this point. We are actually
eager to extend the proposed model and investigate it in a real world setting with real objects and an robotic arm.

**References**

Mike Wu and Noah Goodman. Multimodal generative models for scalable weakly-supervised learning. In *NeurIPS*,
2018.


[Meta-Review · NeurIPS 2019]

An interesting extension of GQN for modelling multimodal representations of 3D scenes using an amortized product of experts to allow training with some of the modalities missing. The paper is not clear enough about prior work on using PoEs for multimodal generative modelling, but the authors promised to correct this. The authors are urged to include the PoE baseline results the reviewers requested when finalizing the paper.